# Physical Fitness Variations between Those Playing More and Those Playing Less Time in the Matches: A Case-Control Study in Youth Soccer Players

**DOI:** 10.3390/children9111786

**Published:** 2022-11-21

**Authors:** Ana Filipa Silva, Filipe Manuel Clemente, César Leão, Rafael Oliveira, Georgian Badicu, Hadi Nobari, Luca Poli, Roberto Carvutto, Gianpiero Greco, Francesco Fischetti, Stefania Cataldi

**Affiliations:** 1Escola Superior Desporto e Lazer, Instituto Politécnico de Viana do Castelo, Rua Escola Industrial e Comercial de Nun’Álvares, 4900-347 Viana do Castelo, Portugal; 2The Research Centre in Sports Sciences, Health Sciences and Human Development (CIDESD), 5001-801 Vila Real, Portugal; 3Research Center in Sports Performance, Recreation, Innovation and Technology (SPRINT), 4960-320 Melgaço, Portugal; 4Instituto de Telecomunicações, Delegação da Covilhã, 1049-001 Lisboa, Portugal; 5Sports Science School of Rio Maior, Polytechnic Institute of Santarém, 2040-413 Rio Maior, Portugal; 6Life Quality Research Centre, 2040-413 Rio Maior, Portugal; 7Department of Physical Education and Special Motricity, Faculty of Physical Education and Mountain Sports, Transilvania University of Braşov, 500068 Braşov, Romania; 8Department of Exercise Physiology, Faculty of Educational Sciences and Psychology, University of Mohaghegh Ardabili, Ardabil 5619911367, Iran; 9Faculty of Sport Sciences, University of Extremadura, 10003 Cáceres, Spain; 10Department of Motor Performance, Faculty of Physical Education and Mountain Sports, Transilvania University of Braşov, 500068 Braşov, Romania; 11Department of Translational Biomedicine and Neuroscience (DiBraiN), University of Study of Bari, 70124 Bari, Italy

**Keywords:** football, athletic performance, youth sports, physical fitness

## Abstract

The purpose of this study was (i) to compare two groups (players with more vs. less match play time) regarding body composition, vertical and horizontal jumping performance, and aerobic capacity; and (ii) to test the relationships between physical fitness and play time. This study followed a case-control design in which the outcome was playtime, and the causal attribute was physical fitness. Sixty-six youth male soccer players from under-16 (n = 21), under-17 (n = 19), under-18 (n = 12), and under-19 (n = 14) age groups were monitored for match play time during five months of observation. Inclusion criteria consisted of (1) no absence of more than a week due to injury or other conditions during the five months of observation and (2) physical assessments having been done simultaneously with those of the other players (at the beginning of the season). The exclusion criteria were (1) not participating in one week or more of training sessions, and (2) not participating in the physical fitness assessments. At the beginning of the season, players were assessed for anthropometry (height, body mass, skinfolds), countermovement jump, triple hop bilateral and unilateral jump, and aerobic capacity using the Yo-Yo Intermittent Recovery Test, level 2 (YYIRT). The group that played more time had significantly greater YYIRT results (+28.2%; *p* = 0.009; Cohen’s d = 0.664). No other significant differences were found between those who played more and fewer minutes. Moderate and significant linear positive correlations were found between YYIRT and play time in the under-19 group (r = 0.423; *p* = 0.031) and overall (r = 0.401; *p* < 0.001). In the case of the under-17 group, moderate and significant linear positive correlations were found between TSA and play time (r = 0.473; *p* = 0.041). This suggests that aerobic and anaerobic capacity is related to play time while jumping performance and fat mass seem not to play an essential role in play time.

## 1. Introduction

Talent identification has challenged researchers and coaches to understand which characteristics achieve the highest levels in soccer [1,2,3,4]. In clubs, there has been an increasing interest in the creation of talent models to scout the most talented youth soccer players at early ages (11–16 years old) [5,6,7,8]. However, defining the concept of talent is not an easy task [9,10]. Traditionally, it has been based on the combination of a precondition for success (e.g., genetic traits) and the outcome of the developmental process (e.g., athletic excellence during youth) [6,8,11]. Therefore, coaches traditionally select young players based on a subjective analysis which leads them to believe that a certain player has the potential to achieve the highest levels of performance [12,13,14]. Evidence of the most common features presented in the greatest soccer players at an early age allows clubs to focus on developing a small number of young players [15,16,17,18].

It is already well known that coaches systematically exclude smaller players, favoring taller ones [19,20,21]. Hence, when comparing different skill levels, the sum of skinfolds, percentage of body fat, and the endomorphic component of somatotype seems to be decisive [22,23]. Anthropometric traits play an important role in achieving the highest performance [24,25]. Furthermore, according to professional soccer data, during a game, the total distance covered is ∼10–12 km at an average work intensity of 80–90% of the maximal heart rate [26,27,28,29]. Therefore, it is unsurprising that the best athletes present greater aerobic and anaerobic fitness [3,30,31]. Indeed, it was also found that successful under-15 and under-16 players have greater aerobic endurance [3,32]. At young ages, explosive strength capacity is also a critical physical factor, determining future contract status [33]. Thoroughly, young soccer players’ physical and physiological characteristics can provide information that is helpful in identifying talented players [25,34,35].

Young athletes from the same chronological age and team can vary significantly in their physical characteristics and capabilities [36]. Typically, at youth levels, coaches choose to play for more extended periods those athletes who exhibit the most advantageous physical characteristics [37,38]. This selection can influence the participants’ exposure to the demands of the match, which may influence physical fitness adaptations, since most of the non-used players will not have a compensatory training stimulus. In addition to the bias of the selection of participants in youth categories, one study conducted in different age groups revealed that the taller, more mature, and more advanced players were given more minutes of play [39]. In the same study, it was revealed that 35% of play time was significantly explained by BMI, maturity offset, lean mass, leg length, and sitting height.

Considering the above background, the aim of the present study was twofold: (i) to compare two groups (players with more vs. less match play time) regarding body composition, vertical and horizontal jumping performance, and aerobic capacity, and (ii) to test the relationships between physical fitness and play time. It was hypothesized that taller soccer players and the athletes with the best performance in physical tests accumulate more minutes of match play time.

## 2. Materials and Methods

### 2.1. Study design

This study followed a case-control design in which the outcome was the play time, and the causal attribute was physical fitness.

### 2.2. Setting

The physical fitness assessment occurred on the following dates: 9 and 10 of August, Monday and Tuesday, respectively, for the under-16 group; 29 and 30 of July, Thursday and Friday, respectively, for the under-17 group; and 19 and 20 of July, Tuesday and Wednesday, respectively, for the under-19 groups. The physical fitness assessment respected 24 hours of rest, in consideration of the previous training session/match. The period of observation regarding the time of play lasted approximately five months. The under-19 group had their first match on 21 August 2021 and the last on 30 December 2021 (a total of 18 matches were observed). The under-17 and under-16 groups had their final matches on the same day, 19 December 2021, while the under-17 group had their first official match on 22 August 2021 with that of the under-16 group being on 3 October 2021. In total, 12 games were observed for the under-16 group and 16 games for the under-17 group. Only official matches of the principal championship in which they were enrolled were considered in determining the total playing time. The time of play was recorded in minutes, and the sum of all periods over the observation period was determined for each player.

### 2.3. Participants

Eighty-seven youth male soccer players (age: 16.50 ± 1.08 years; height: 1.74 ± 0.07 m; body mass: 66.86 ± 8.34) from under-16 (number of participants = 29), under-17 (number of participants = 28) and under-19 (number of participants = 30) age groups were monitored for match play time during 5 months of observation. Inclusion criteria consisted of (1) no absence for more than a week due to injury or other conditions during the five months of observation, and (2) physical assessments having been done simultaneously with those of the other players (at the beginning of the season). The exclusion criteria were (1) not participating in one week or more of training sessions, and (2) not participating in the physical fitness assessments. Given the eligibility criteria, after the recruitment period, sixty-six youth male soccer players (16.56 ± 1.14 years old; 1.75 ± 0.07 m; 66.75 ± 7.89 kg) from the under-16 (n = 21), under-17 (n = 19), and under-19 (n = 26) age groups were included in our analysis. Table 1 presents the players’ anthropometric details. The study design, benefits, and possible risks were verbally presented to the players and their parents/guardians. After their agreement, a written informed-consent document was signed by both parties before testing. All tests were conducted in the first week of training for every age group in the season 2021–2022 (under-19: 19 to 20 of July; under-17: 29–30 August; under-16: 9–10 August). The design was conducted in accordance with the ethical guidelines of the Declaration of Helsinki for the study of humans. The Escola Superior de Desporto e Lazer ethical committee approved the study with the code CTC-ESDL-CE001-2021.

### 2.4. Data Collection

The anthropometry and physical assessment occurred over two days. The assessments were performed on day one by 3:30 p.m., starting with anthropometry and followed by vertical and horizontal jumps. The Yo-Yo Intermittent Recovery Test was performed the day after. The anthropometry and the vertical and horizontal jumps were performed in a room with a temperature of 21 °C, and relative humidity of around 40%. The remaining tests were performed on a synthetic turf, with an average temperature between 22 and 21 °C on 19 and 20 July; 20 and 21 °C on 29 and 30 of July; and 22 and 23 °C on 9 and 10 August. On all the occasions, the weather was clear, without rain. The rest of the tests allowed 1 min between jumps. YYIRT was performed on the day after the anthropometry test and vertical and horizontal jumps were performed.

### 2.5. Anthropometry

The anthropometric assessment was conducted in a room with a temperature of 23 °C. The players were instructed to wear light clothing and stood barefoot. A portable stadiometer was used to measure the players’ statures (Seca 217, Hamburg, Germany; error nearest 0.1 cm). A digital scale was used to measure body mass (Prozis SmartScale; error nearest 0.1 cm). The anthropometric measures also included the assessment of eight skinfolds: triceps, biceps, subscapular, abdominal, suprailiac, supraspinal, calf, and thigh. The skinfolds were assessed twice (at 0.1 mm). A Harpenden caliper (British Indicators, Ltd., London, UK) was used to perform the assessments. The mean value of the two measurements was recorded and the sum of eight skinfolds (8SKF) calculated. To estimate fat mass, the equation of Reilly [40] was applied. Fat mass in percentage was considered the outcome for data analysis. Furthermore, muscle mass was estimated using the equation of Lee [41]. The anthropometric procedures were in line with the guidelines of the International Society for the Advancement of Kinanthopometry [42].

### 2.6. Vertical Jumps

The countermovement jump (CMJ) was implemented as a vertical jump test. All the participants performed the test at the same time of the day (in the afternoon), immediately before the training sessions. Before the assessments, a standardized warm-up (consisting of low-to-moderate running and dynamic stretching of the lower limbs) was performed. The CMJ began in the upright position with the arms on the waist [43]. After a verbal cue was given, the players flexed their knees and jumped as high as possible. It was requested that they jump with their legs straight and perform plantar flexion. Three repetitions of CMJ were performed, interspaced by 60 seconds of rest [44]. A contact platform was used for measuring the vertical jump (Chronojump) [45]. The contact platform was attached to hardware (Chronopic^®^, Chronojump Boscosystem, Barcelona, Spain). Free software (Chronojump Boscosystem Software, Spain) was used to extract information regarding the jump’s height. The highest jump (from three) was recorded for further data treatment [46]. The final outcome obtained was CMJ height (cm).

### 2.7. Horizontal Jumps

Both single-leg triple hop jump (SLTH) and bilateral triple hop jump (BTH) were performed by each participant. An open area was prepared, with a measuring tape of at least 8 m taped on the floor. A strip was placed perpendicular to this tape to create the starting line.

For the SLTH test, we instructed the athletes to begin by standing on a selected leg and position themselves immediately before the start line. They were asked to perform three consecutive maximal hops and to land with the preferred leg. A measuring tape measured the distance between the start line and the athlete’s heel. The participants performed three trials with each leg. Each leg was given 60 seconds of rest between repetitions.

For the case of BTH, the players started with both feet in parallel position before the start line. After a verbal cue, they performed a triple jump with no balance run. A measuring tape measured the distance between the start line and the athlete’s heel. The participants performed three trials, interspaced by 60 seconds of rest.

The best performances obtained in the right and left SLTH and in the BTH were used for data treatment.

### 2.8. Yo-Yo Intermittent Recovery Level 2 (YYIRT)

The players performed the Yo-Yo Intermittent Recovery Test, level 2. The test consists in performing two 20 m runs at progressively increasing speeds while recovering for 10-s after every two rounds [47]. The methodological procedures of the test can be observed in the original study of the validation and reliability of the test [48]. An audio beep governed the pace. The players’ performance was obtained from the last round in which each player completed the run. The total distance was collected as the final outcome.

### 2.9. Total Score of Athleticism (TSA)

Each physical fitness outcome was standardized using a z-score with the specific equation of the value of the player subtracted by the average of the group for the outcome, and the result was divided by the standard deviation of the group. The TSA [49] was calculated based on the sum of the z-scores of the CMJ, TH bilateral, TH right leg, TH left leg, and YYIRT.

### 2.10. Independent Variables

An independent variable was created based on the time of play (measured in minutes and representing the sum of minutes accumulated in the matches considered during the study period). The groups having less or more play time were defined as those below or above the median time for each age group, respectively. In this case, the median time of play for the under-16, under-17, and under-19 groups were 531, 407 and 486 min, respectively. In the case of the under-16 group, ten players were included in the group with less play time, while eleven were included in the group with more play time. In the case of the under-17 group, nine players were included in the group with less play time, while ten were included in the group with more play time. Finally, in the case of the under-19 group, thirteen players were included in the group with less play time, while thirteen were included in the group with more play time.

### 2.11. Statistical Procedures

A preliminary inspection of outliers was executed for main physical fitness outcomes. No significant outliers were found. The normality and homogeneity of the physical fitness outcomes were then inspected using Kolmogorov-Smirnov and Levene’s tests, respectively. The tests revealed the normality (*p* > 0.05) and homogeneity (*p* > 0.05) of the sample for all physical fitness outcomes. A one-way ANOVA was executed to compare the primary outcomes between age groups. Eta squared was calculated as effect size, and the Bonferroni test was executed as a post hoc test. An independent t-test was executed to test variations of play time between those who had less and more play time. The standardized effect size of Cohen (Cohen’s d) was calculated, since the pooled standard deviation was used as part of the equation. Pearson’s product-moment correlation test was executed to test the relationships between physical fitness outcomes and play time. The following thresholds were used to characterize the magnitude of correlations [50]: 0.0–0.1, trivial; 0.1–0.3, small; 0.3–0.5, moderate; 0.5–0.7, large; 0.7–0.9, very large; and 0.9–1.0, nearly perfect. The statistical tests were computed in SPSS software (version 28.0.0.0., IBM, Chicago, IL, USA) for a *p* < 0.05.

## 3. Results

Descriptive statistics of body fat and play time (measured in minutes and representing the sum of minutes accumulated in the matches considered during the study period) for each age group can be observed in Figure 1. The under-16 group presented a greater average of body fat (11.3%), while a greater average of play time was observed in the under-19 group (656 minutes). No significant differences in body fat were found between age groups (*p* = 0.631; $\eta^{2}$ = 0.016). Additionally, no significant differences in time of play were found between age groups (*p* = 0.790; $\eta^{2}$ = 0.008).

Descriptive statistics of triple hop and countermovement jump for each age group can be observed in Figure 2. The under-19 group presented the highest averages in triple hop bilateral (7.4 m), in triple hop left and right legs (6.7 m), and in CMJ (40 cm). Significant differences between age groups were found in triple hop right leg (*p* < 0.001; $\eta^{2}$ = 0.373) and left leg (*p* < 0.001; $\eta^{2}$ = 0.245); triple hop bilateral (*p* < 0.001; $\eta^{2}$ = 0.523); and countermovement jump (*p* < 0.001; $\eta^{2}$ = 0.487).

Descriptive statistics of the YYIRT for each age group can be observed in Figure 3. The under-19 group presented greater average results (491 m). Significant differences between age groups were found in the YYIRT results (*p* < 0.001; $\eta^{2}$ = 0.361).

Table 2 presents the descriptive statistics of physical fitness outcomes for the groups who had less play time and more play time in each age group. Table 2 presents the overall result for all age groups considered. The group of players with more accumulated minutes of play had significantly greater YYIRT results (+28.2%; df: 64; t = −3.681; *p* = 0.009; Cohen’s d = 0.664).

Table 3 presents the matrix of correlations between play time and the physical fitness outcomes considered in the current study. Moderate and significant linear positive correlations were found between YYIRT results and play time in the under-19 group (r = 0.423; *p* = 0.031) and overall (r = 0.401; *p* < 0.001). In the case of the under-17 group, we found moderate and significant linear positive correlations between TSA and play time (r = 0.473; *p* = 0.041).

## 4. Discussion

The current research results revealed that body fat was lower in the oldest group. At the same time, play time, unilateral triple hop, bilateral triple hop, countermovement jump, and YYIRT results presented higher values in the older groups. These results were proportional to the three age-group categories.

Considering the study’s first aim, only two differences between the groups were found for YYIRT results in both meters (distance covered) and z-score values, with higher values found in the group with more play time. The YYIRT analyses the ability to repeatedly perform an intermittent exercise in which aerobic and anaerobic energy systems are used [51]. This test was chosen because soccer is also characterized by the same physiological characteristics, which means that soccer players depend on well-developed energy systems to respond to the different types of actions in the match, such as running at a moderate speed (aerobic system) or jumping, sprinting, and/or scoring a goal (anaerobic system) [52]. Moreover, higher performance in this test is associated with lower risk of injury [53]. This information helps to explain the observation that the athletes with more play time also presented greater cardiorespiratory capacity.

With respect to the second aim of this study, we also confirmed the previous result by showing an association between YYIRT results and play time in the under-19 group and among the overall participants. This was not shown by the other two age-group categories, which led to speculation that older athletes with higher levels of YYIRT performance contributed more to the overall results and that in older athletes, this capacity has a major role when compared to younger athletes.

Contrary to the present results, one study found that standing broad jump performance significantly predicts play time in young soccer players (16 years) [33]. Although the present study did not use the test employed in that study, we used unilateral and bilateral TH tests but failed to show any correlation. The TH test is not sufficiently sensitive to determine play time, although this could be confirmed in future studies.

Furthermore, the study of Deprez et al. conducted a deep analysis of play time. From a group of 29, six players participated in more than 50% of all matches from the season [33]. Additionally, it was found that these six players presented more explosive power (standing broad jump) and were older than players with fewer minutes of match play (19.4 vs. 18.6 years) [33].

Using a different approach to analyzing younger players (under 11–14 categories), Clemente et al. tested the influence of anthropometry, body composition, and maturity offset on play time, with results explaining 35% of the variation in play time [39]. Other studies found that body fat percentage was a factor in selecting players when compared to non-selected players [54]. Along the same line, higher values of lean body mass seem to contribute to better jumping and sprinting performance [55]. Nonetheless, in younger categories, anthropometric and body composition variables showed higher relevancy to determining play time [39,54], which was not the case in the present study.

Another relevant finding from this study was related to the moderate association between TSA and play time in the under-17 group. In the present study, the TSA derived from all tests, including body fat and CMJ, bilateral and unilateral Triple Hop bilateral, and the YYIRT. However, it is essential to highlight that there needs to be a consensus on the tests best suited to define TSA [56].

Our results support the conclusion that the older group (under-19) presented the best results for all tests, while lower values were observed in the younger categories. However, TSA showed an association only in the under-17 group, which led us to speculate that the tests used to calculate TSA are more accurate for this age group. Even so, this speculation needs further study with more teams and larger sample sizes.

Apart from the YYIRT, which has already been explained, body fat, CMJ, and bilateral and unilateral TH tests were applied to access several capabilities in soccer players, since this sport has multiple components, including including body composition, strength, and conditioning variables [57,58,59,60]. All these tests possess specific characteristics that can add to the determination of the play time of a player [33,39]. For instance, one study showed that players with better anaerobic, vertical power, and sprint performance were selected for international/professional soccer teams [35]. Corroborating the previous results, it was shown that non-specific soccer motor coordination tests play a major role in identifying Belgian international young soccer players (15–16 years) [57].

### Study Limitations

This study presents some limitations: (a) only tests related to body composition (skinfolds), jumping performance (TH and CMJ), and cardiorespiratory (YYIRT) capabilities were performed, while future studies should include other non-specific and specific soccer field tests; (b) similarly, the calculation of TSA could reveal different results if other tests were being used; (c) the sample size was relatively small considering each age group; and d) only five months were used to collect play time, which could influence the results with higher or lower follow-up. Therefore, future studies should include other tests and instruments/equipment to assess the different capabilities of players. In addition, a larger sample size and a full-season study should be considered to address limits regarding play time. In this last scenario, we recommend performing more assessments throughout the season.

Despite these limitations, to the best of our knowledge, this study is the first to use this methodology and to identify the relevancy of the YYIRT to play time. It helps coaches and their staff better understand that, with age, aerobic and anaerobic assessments are warranted. Although the other results were insignificant, assessments from all types should be performed for better training monitoring.

## 5. Conclusions

The current research revealed that the players participating in more minutes of play time presented greater aerobic fitness than those with fewer minutes in the game. The results were irrespective of the age group. However, no significant differences were found between players with more or fewer minutes of play regarding the triple hop jump test and countermovement jump. A similar absence of differences was found regarding body composition. Comparisons between age groups also revealed that the older, the better, in physical fitness measures. Despite the small sample size and other study limitations, this research indicates that aerobic fitness can be highly correlated with participation and match in this context. Coaches must pay special attention to this, aiming to balance the training process.

## Figures and Tables

**Figure 1 children-09-01786-f001:**
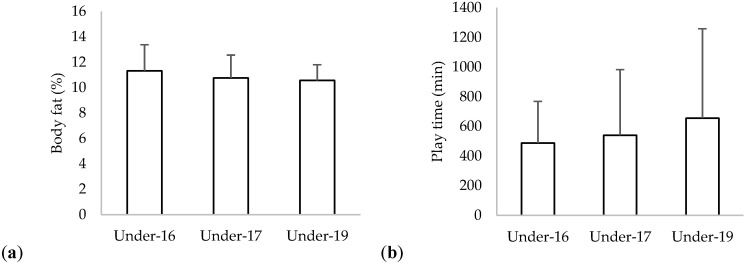
Descriptive statistics (mean and standard-deviation) of (**a**) body fat and (**b**) time of play.

**Figure 2 children-09-01786-f002:**
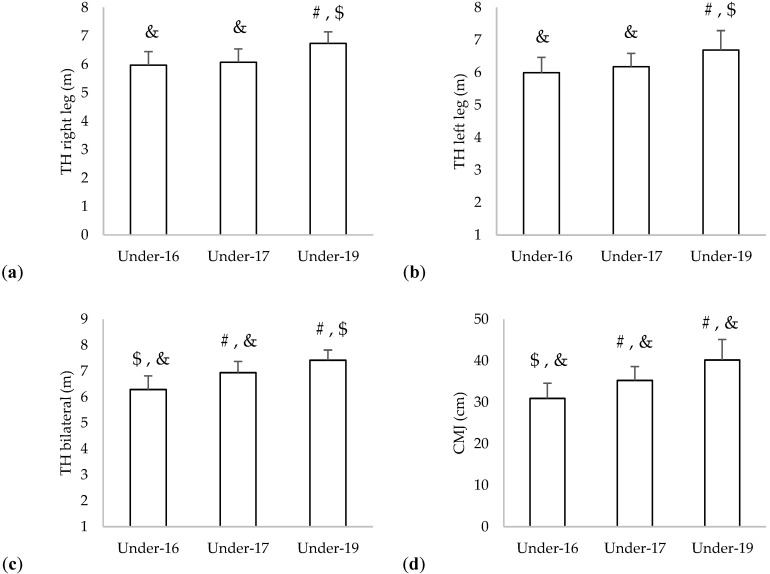
Descriptive statistics (mean and standard deviation) of the (**a**) triple hop right leg and (**b**) triple hop left leg; (**c**) triple hop bilateral; and (**c**) countermovement jump. #: significantly different (*p* > 0.05) from under-16; $: significantly different (*p* > 0.05) from under-17; &: significantly different (*p* > 0.05) from under-19.

**Figure 3 children-09-01786-f003:**
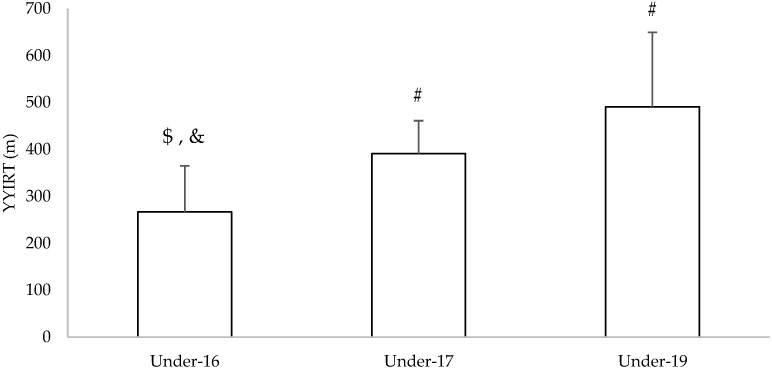
Descriptive statistics (mean and standard deviation) of the Yo-Yo Intermittent Recovery Test. #: significantly different (*p* > 0.05) from under-16; $: significantly different (*p* > 0.05) from under-17; &: significantly different (*p* > 0.05) from under-19.

**Table 1 children-09-01786-t001:** Mean and standard deviation of demographic and anthropometric measures for each age group.

	All Athletes (n = 87)			Athletes Included in Analysis (n = 66)		
	U-16 (n = 29)	U-17 (n = 28)	U-19 (n = 30)	U-16 (n = 21)	U-17 (n = 19)	U-19 (n = 26)
Age (years)	15.4 ± 0.3	16.3 ± 0.2	17.8 ± 0.6	15.4 ± 0.4	16.3 ± 0.3	17.8 ± 0.7
Height (m)	1.7 ± 0.1	1.8 ± 0.1	1.8 ± 0.1	1.7 ± 0.1	1.7 ± 0.1	1.8 ± 0.1
Body mass (kg)	61.6 ± 7.8	67.7 ± 7.4	71.1 ± 6.8	61.6 ± 8.0	65.8 ± 5.6	71.6 ± 6.3
BMI (kg.m^2^)	21.1 ± 2.0	22.1 ± 0.9	22.6 ± 1.4	21.3 ± 1.9	21.6 ± 1.7	22.5 ± 1.4
Skinfold Sum (mm)	71.0 ± 22.1	73.0 ± 22.7	67.7 ± 12.3	75.8 ± 23.8	72.2 ± 24.2	67.7 ± 13.0

BMI: body mass index; U: under.

**Table 2 children-09-01786-t002:** Descriptive statistics (mean and standard deviation) of physical fitness outcomes for those with less and more play time.

	Lower Time of Play	Higher Time of Play	% Difference (Higher–Lower)	t	*p*	Cohen’s d (Higher–Lower)
Body fat (%)	11.0 ± 1.8	10.8 ± 1.7	−1.8	0.458	0.649	0.113
CMJ (cm)	35.5 ± 6.3	36.2 ± 5.0	2.0	−0.506	0.615	−0.125
TH bilateral (m)	6.9 ± 0.7	6.9 ± 0.6	0.0	−0.105	0.917	−0.026
TH right leg (m)	6.2 ± 0.6	6.4 ± 0.5	3.2	−1.127	0.264	−0.277
TH left leg (m)	6.3 ± 0.6	6.3 ± 0.5	0.0	−0.225	0.823	−0.055
YYIRT (m)	341.3 ± 127.3	437.4 ± 159.4	28.2	−2.696	0.009 *	−0.664
Body fat z-score	0.0 ± 1.1	−0.2 ± 1.0	−700	0.775	0.441	0.191
CMJ z-score	−0.1 ± 1.2	0.2 ± 0.9	−300	−0.963	0.339	−0.237
TH bilateral z-score	0.0 ± 1.1	0.1 ± 0.9	100	−0.110	0.913	−0.027
TH right leg z-score	−0.3 ± 1.1	0.1 ± 0.8	−133.3	−1.633	0.107	−0.402
TH left leg z-score	−0.2 ± 1.0	0.0 ± 1.0	−100.0	−0.512	0.611	−0.126
YYIRT z-score	−0.4 ± 0.9	0.4 ± 0.9	−200.0	−3.681	<0.001 **	−0.907
TSA (A.U.)	−0.8 ± 3.9	0.7 ± 3.9	−187.5	−1.844	0.070	−0.454

df: degrees of freedom; CMJ: countermovement jump; TH: triple hop; TSA: total score of athleticism; * significant different at *p* < 0.05; ** significant different at *p* < 0.01.

**Table 3 children-09-01786-t003:** Correlation table between time of play (min) with the remaining physical fitness outcomes.

	BF	CMJ	TH Bilateral	TH Right Leg	TH Left Leg	YYIRT	TSA
Under-16	0.035 [−0.403;0.460]	0.176 [−0.277;0.565]	−0.076 [−0.492;0.368]	0.158 [−0.293;0.552]	0.234 [−0.219;0.605]	0.359 [−0.086;0.684]	0.202 [−0.251;0.583]
Under-17	−0.332 [−0.683;0.144]	0.352 [−0.122;0.695]	0.386 [−0.083;0.715]	0.219 [−0.262;0.612]	0.041 [−0.421;0.486]	0.402 [−0.064;0.724]	0.473 * [0.024;0.723]
Under-19	−0.141 [−0.501;0.261]	−0.017 [−0.401;0.373]	−0.235 [−0.570;0.168]	0.048 [0.346;0.428]	−0.104 [−0.473;0.295]	0.423 * [0.042;0.696]	0.021 [−0.370;0.405]
Overall	−0.161 [−0.388;0.084]	0.174 [−0.071;0.399]	0.095 [−0.151;0.329]	0.184 [−0.061;0.408]	0.071 [−0.174;0.307]	0.401 ** [0.176;0.586]	0.168 [−0.078;0.394]

BF: body fat; CMJ: countermovement jump; TH: triple hop; YYIRT: yo-yo intermittent recovery test; TSA: total score of athleticism; * significant at *p* < 0.05 ** significant at *p* < 0.01.

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
