# Peer review of "Physical Fitness Variations between Those Playing More and Those Playing Less Time in the Matches: A Case-Control Study in Youth Soccer Players"

_children, 2022, doi:10.3390/children9111786_

Round 1
Reviewer 1 Report
1. In this study, it is not known whether good physical fitness affects play time or whether the length of play time affects the improvement of physical fitness. This title is inappropriate because the causal relationship is unknown. You can only speak about association. Similar wording in the text should also be corrected.
2. Please indicate the gender of the participant.
3. Is the order of 531, 407, and 486 in Line 215 correct?
4. (Lines 211-215) Please list the number of Lower and Higher students in each age group.
5. The text and units in the figures are too small to read.
6. Figures 1, 2, and 3 should be compared age groups and significant items should be marked with an asterisk or similar.
7. Bar graphs should generally contain SD or SE bars, not plots.
8. Please reconsider whether vertical lines are necessary for tables, and check the journal's regulations.
Reviewer 2 Report
Thank you for preparing the article. Please make the correction.
Thank you for preparing the article.
1. Please state what the purpose of the work was.
2. What the inclusion and exclusion criteria were for inclusion in/exclusion from the study?
3. Line 109 - lack of information that the subjects and their parents/guardians signed an informed consent for participation in the study.
4. Line 267 - What do you mean they played more? What unit of time is that? "Those played more time...."
Please correct such a statement throughout the manuscript - it does not sound scientific.
5. Line 268: t = –3.681, please explain why the authors refer to this value.
6. Limitations - please post as a separate item after the discussion.
7. Line 290: "The aim of this study was..." this should be in the introduction.
8. Conclusion - please improve by referring to all the results obtained.
Round 2
Reviewer 1 Report
Thank you for your response of the revision. We accept the revision.
Reviewer 2 Report
Congratulations to the authors for the research idea and the preparation of the publication, taking into account the reviewers' comments.
